# An Explainable AI-Based Decision Support Tool to Predict Preterm Birth

Ilias Kyparissidis-Kokkinidis,
Emmanouil S. Rigas,
Evangelos Logaras
Lab of Medical Physics
and Digital Innovation,
School of Medicine,
Aristotle University of Thessaloniki,
Greece
Email: {iliaskypkok, erigas,
evanlogar}@auth.gr

Ioannis Tsakiridis,
Themistoklis Dagklis
3rd Department of
Obstetrics and Gynecology,
School of Medicine,
Aristotle University of Thessaloniki,
Greece
Email: {igtsakir, dagklis}@auth.gr

Paraskevas Lagakis,
Antonis Billis,
and Panagiotis D. Bamidis
Lab of Medical Physics
and Digital Innovation,
School of Medicine,
Aristotle University of Thessaloniki,
Greece
Email: {plaga, ampillis,
bamidis}@auth.gr

*Abstract*—**Preterm Birth (PTB), characterized as birth occurring prior to 37 weeks of gestation, presents a notable clinical challenge. In this work, we aim to assist the decision making process of the obstetricians by proposing an AI-based clinical decision support system. Specifically, we propose a Machine Learning (ML)-based model to efficiently predict PTB using an assortment of relevant features including social demographics, medical history, along with laboratory and obstetric examination results. This model was trained and validated using a dataset consisting of $873$ women from a major Greek hospital. Moreover, we implemented an explainability feature using SHapley Additive exPlanations (SHAP) to enhance the clinical interpretation of the results. Additionally, we developed a web application that encompasses both the predictive model and the explainability feature in an easy-to-use user interface. The predictive model has shown strong performance in internal validation, achieving an accuracy of $94\%$ and a recall of $97\%$. In external validation, where the tool was used by clinicians for 100 pregnant women, it achieved an accuracy of $89\%$ and a recall of $94.3\%$. Finally, the web application was well accepted by the clinicians.**

*Index Terms*—**Artificial Intelligence, AI, Machine Learning, ML, Explainable Artificial Intelligence, XAI, Premature Birth, preterm birth, Obstetrics, Gynecology, Clinical Decision Support Systems, Clinical**

## I. INTRODUCTION

The integration of artificial intelligence (AI) into the healthcare sector creates a new pathway towards more predictive, personalized, and efficient medical practices [1]. One promising application of AI in this domain is the prediction of preterm birth, a condition that significantly impacts neonatal outcomes and places considerable emotional and financial burdens on families and healthcare systems. Preterm birth, defined as delivery before 37 weeks of gestation, is a leading cause of neonatal mortality and morbidity, making its prediction and prevention a critical focus in obstetric care [2].

Digital health [3], encompassing a broad range of technologies facilitate the continuous collection of large datasets, which are essential for training AI models. In the context of preterm birth prediction, AI models can analyze several factors to identify high-risk pregnancies with greater accuracy and at earlier stages than conventional methods [4]. AI's role in medicine extends to enhancing clinical decision support systems (CDSS), which are designed to aid healthcare providers in making informed clinical decisions [5]. By integrating AI into CDSS, healthcare providers can benefit from evidence-based insights and predictive analytics, thereby improving the quality of care and citizens' health outcomes. For preterm birth, AI-enhanced CDSS can offer timely risk assessments, recommend personalized interventions, and support clinicians in monitoring and managing high-risk pregnancies. This enhances the precision of prenatal care and optimizes resource allocation and reduces unnecessary interventions.

In this work, we focus on the problem of predicting pre-term birth, for women at the $28^{th}$ week of gestation and later. To do this, we used a dataset consisting of demographics, medical history, blood tests and pregnancy related exams and we built a machine learning model to make the prediction. To select the best performing model, several state-of-the-art ML algorithms have been tested. On top, we employed an explainable AI feature using SHAP [6], in order to enhance interpretability of the results. We also developed a web application that incorporates the ML model and the explainability feature in an easy to use interface. To train our algorithms, a dataset consisting of 837 pregnant women from Hippokration General Hospital of Thessaloniki Greece was used, whereas the tool was validated in a real-world feasibility study in the same hospital where it was used in parallel to the established medical practice for 100 pregnant women. This work extends our previous work [7] which presented preliminary results of the effort to predict preterm birth using a dataset of 375 cases. In that work only an internal validation of a set of ML models took place, and a basic explainability feature was provided. The results were promising, but inferior to the ones presented in the current work.

The rest of the paper is structured as follows: Section II presents related works and outlines the novelty of this work.

Section III describes the considered algorithms for the predictive model, the explainability algorithm and the results of the internal validation of the predictive models. Section IV describes the web application and its high-level architecture. Section V presents the results of the external validation of the predictive model and the tool. Finally, Section VI concludes and presents ideas for future work.

## II. RELATED WORK

In terms of predicting preterm birth, some interesting works exist. Koivu and Sairanen [8] utilize a vast dataset from the Centers for Disease Control and Prevention (CDC) comprising almost sixteen million observations to develop models predicting early stillbirth, late stillbirth, and preterm birth. They apply logistic regression, artificial neural networks, and gradient boosting decision trees, achieving AUC scores of $0.76$ for early stillbirth, $0.63$ for late stillbirth, and $0.64$ for PTB, highlighting the importance of extensive data and complex models for robust predictions. Watson et al. [9] evaluate the effectiveness of the QUiPP App, which uses a predictive model combining history of spontaneous preterm birth (sPTB), gestational age, and quantitative measurements of fetal fibronectin (qfFN) to triage women at risk of sPTB. The study found that using the QUiPP App at a $5\%$ risk threshold for intervention could correctly identify all true cases of preterm labor while potentially avoiding unnecessary hospitalizations and interventions for $89.4\%$ of women presenting before 30 weeks' gestation. Moreira et al. [10] propose a machine learning technique using the support vector machine (SVM) algorithm to predict preterm birth risk in a pregnancy database. They highlight the superior performance of SVM over other machine learning methods, achieving an accuracy of $0.821$, a true positive rate of $0.839$, and a receiver operating characteristic (ROC) area of $0.785$. The model is designed for integration into mobile health applications, providing decision support to healthcare providers by predicting preterm birth risk anytime and anywhere. Finally, AlSaad et al. [11] introduce a clinical prediction model utilizing recurrent neural networks (RNNs) with a single code-level attention mechanism to predict preterm birth (PTB) at various time points before delivery. Leveraging a large dataset of 222.436 deliveries, the model achieved ROC-AUC scores of $0.82$, $0.79$, and $0.78$ for predictions made at 1, 3, and 6 months prior to delivery, respectively. The attention mechanism also provides temporal explanations for clinicians, improving model interpretability.

Compared to the previous works, our study also employs machine learning for PTB prediction, and demonstrates equal or superior performance with higher accuracy and recall rates in both internal (see Section III-D) and external (see Section V - that took place in Hippokration General Hospital of Thessaloniki along with the established clinical practice) validations and includes the development of a web-based application for clinical routine use, emphasizing usability and transparency in predictions using an explainable AI (XAI) technique.

## III. METHODS

In this section, the methodological approach that was followed to collect the data, pre-process them and to train and test the predictive model is described.

### A. Data collection

For the development of a robust preterm birth prediction model, an extensive dataset comprising 873 pregnancies with gestational age greater than or equal to 28 weeks was employed, with 293 cases identified as preterm. This dataset, encompassing 24 features including demographics, social and medical history, and obstetric variables, was collected as part of an ongoing prospective cohort study conducted at the Hippokration General Hospital. We consulted clinicians for the selection of the predictor variables/parameters and all collected data were timestamped. The data collection process spanned four months (May-August 2022) and was carried out by a team of four medical professionals. This initiative was embedded within the framework of the HosmartAI project[1] and received ethical approval from the Research Ethics and Conduct Committee of the Aristotle University of Thessaloniki (AUTH) under protocol number 94521/2022. All information was collected pseudonymously to maintain patient confidentiality while ensuring data integrity and usability.

### B. Dataset Characteristics and Preprocessing

The collected dataset included a diverse array of features categorized into numerical, ordinal, and nominal types. Key variables encompassed age, BMI, smoking status, medical history, gravidity, parity, and several obstetric parameters. To prepare the dataset for model development, a thorough preprocessing phase was implemented, involving several steps: 1) **Categorization and Encoding**: Features were categorized based on their type (numerical, ordinal, or nominal). One-hot encoding was utilized for nominal variables, transforming categorical data into binary vectors. Ordinal variables were mapped using a label encoder to convert them into integer values. 2) **Handling Missing Data**: Features with high levels of missing data were excluded from the dataset to maintain data quality. Missing values in remaining features were imputed using either the most frequent value (for categorical data), or the median value (for numerical data). 3) **Balancing the Dataset**: To address class imbalance, random under-sampling of the majority class and the Synthetic Minority Oversampling Technique (SMOTE) [12] were applied. This ensured that the model training phase had a balanced representation of preterm and term birth cases, enhancing the model's ability to generalize across different scenarios. Detailed baseline characteristics of the development data as well as the dataset's variables are shown in Tables I and II.

### C. Predictive models

To provide accurate predictions for preterm birth, a variety of machine learning models were tested. These models

---

[1]https://www.hosmartai.eu/

| Variable | No | Yes |
|---|---|---|
| Smoking | 788 (90.26%) | 85 (9.74%) |
| Conception_ART | 794 (90.95%) | 79 (9.05%) |
| History_of_SGA_FGR | 862 (98.74%) | 11 (1.26%) |
| History_of_PTD | 850 (97.37%) | 23 (2.63%) |
| Preterm_birth | 580 (66.44%) | 293 (33.56%) |
| Gravida | 374 (42.84%) | 499 (Various 1-7) (57.16%) |
| Parity | 530 (60.71%) | 344 (Various 1-6) (39.29%) |
| Placental Cord Insertion | 220 (25.2%) | 726 (Various 1-3) (74.8%) |
| Placental Cord Insertion Abnormal | 824 (94.38%) | 49 (5.62%) |
| Placental Location | N/A | Observations (Various Locations) |
| Single Umbilical Artery | 867 (99.31%) | 6 (0.69%) |
| Maternal Disease | 704 (80.64%) | 169 Total (1's from all diseases) (19.36% calculated separately) |

TABLE I
LIST OF CATEGORICAL VARIABLES AND THEIR DISTRIBUTION INCLUDED IN THE DATASET.

| Variable | Mean | Std. Dev |
|---|---|---|
| Maternal age | 32.53 | 5.16 |
| UtA doppler | 0.81 | 0.23 |
| b-hcg | 1.00 | 0.29 |
| DVP | 4.67 | 0.87 |
| MCA doppler | 2.01 | 0.23 |
| Papp-A | 0.91 | 0.26 |
| Height | 165.66 | 6.11 |
| UA doppler | 0.94 | 0.15 |
| DV doppler | 0.61 | 0.18 |
| EFW | 1905.98 | 267 |
| Cervical length | 27.45 | 8.12 |
| BMI | 28.06 | 4.98 |

TABLE II
LIST OF THE NUMERICAL VARIABLES INCLUDING THEIR MEAN VALUES AND STANDARD DEVIATIONS.

included: i) Random Forest [13], an ensemble method that constructs multiple decision trees and merges their outcomes to improve accuracy and control over-fitting. Random Forests are robust to overfitting and can handle large datasets with higher dimensionality, providing a good balance between bias and variance. ii) Support Vector Machine (SVM) [14], a non-linear model that uses a kernel trick to handle complex data distributions. SVMs are effective in high-dimensional spaces and are memory efficient, making them suitable for the diverse and complex nature of obstetric data. iii) Logistic Regression [15], a linear model used for binary classification tasks, providing probability outputs for predictions. It was chosen for its simplicity, and innate interpretability. iv) K-Nearest Neighbors (KNN) [16], a simple, instance-based learning algorithm that classifies a data point based on the majority class of its nearest neighbors. KNN is intuitive and easy to implement, providing a baseline to compare more complex models against. v) Decision Tree [17], a model that uses a tree-like structure of decisions and their possible consequences to classify data points. Decision Trees are simple to understand and interpret, making them useful for initial explorations of the dataset. vi) Gradient Boosting [18], an ensemble technique that builds models sequentially, with each new model correcting errors made by previous ones. Gradient

Boosting is powerful for its ability to improve model accuracy through iterative corrections, making it suitable for complex prediction tasks. vii) AdaBoost [19], an ensemble method that combines multiple weak classifiers to create a strong classifier by focusing on the hardest to classify instances. AdaBoost enhances the performance of simple models by adaptively adjusting to the data's nuances. viii) XGBoost [20], an advanced gradient boosting algorithm known for its efficiency and performance on structured data. XGBoost is particularly powerful for handling missing data and is highly customizable through its many hyperparameters, which is ideal for optimizing model performance. ix) Naive Bayes [21], a probabilistic classifier based on Bayes' theorem, assuming independence between predictors. Naive Bayes is highly efficient and performs well with small datasets and categorical input features. x) Bagging [22] (Bootstrap Aggregating [23]), an ensemble method that improves the stability and accuracy of machine learning algorithms by training multiple models on different subsets of the data. Bagging reduces variance and helps in preventing overfitting. xi) Model Ensemble Methods [24], to further enhance model performance, various combinations of Voting and Stacking ensembles [25] were tested. Voting involves aggregating the predictions of multiple models, leveraging their collective strengths to improve overall prediction accuracy. Stacking uses the predictions of several models as input to a higher-level model, which makes the final prediction. This method effectively combines the individual strengths of various models to produce more accurate and robust predictions. The goal was to identify the model that offered the best performance in predicting preterm births, ensuring reliable and actionable outputs for clinical use.

*1) Hyperparameter Optimization:* Hyperparameters [26] for each model were fine-tuned using Bayesian optimization [27], a probabilistic model-based approach for finding the optimal parameters. Bayesian optimization was chosen for its efficiency in navigating the hyperparameter space, balancing exploration (trying out new, potentially better configurations) and exploitation (refining known good configurations) to find the best model settings. This method leads to better model performance by systematically searching for the best combination of hyperparameters, rather than relying on trial-and-error or grid search methods. Additionally, error analysis was conducted to identify and rectify any inconsistencies or biases in the models. By analysing where the models made incorrect predictions, adjustments were made to improve their accuracy and reliability. This approach ensured that the final models were not only optimized for accuracy, but also for high sensitivity to provide robust and dependable predictions for preterm births in a clinical setting.

*2) Model Interpretation Using SHAP Explanations:* To provide an interpretable explanation to clinicians, feature importance is calculated using python's SHAP library which can explain any ML-model output using Shapley values from game theory. This gives any machine learning model's output a game-theoretic explanation. Using the common Shapley values from game theory and their accompanying expansions,

it draws a link between optimal credit allocation and local explanations. Model independence is a feature of SHAP values, which can explain both generally for any model and locally for each prediction. This is especially helpful when it comes to clinical practice because clinicians, who are typically cautious in their professional judgment, are more likely to accept an interpretable model than a black-box one.

*D. Internal validation*

To provide the prediction for PTB, the models shown in Table III were implemented and tested using the scikit learn implementation [28]. The final model selected for use in the application is the Optimized Voting which is a combination of the best performing iterations of Logistic regression and XGBoost. As mentioned earlier, this study is focused on women at the 28th week of gestation or later. All predictive variables are available at this point in time. Thus, our model can predict preterm delivery up to 9 weeks ahead.

Using the timestamps we managed to cross-check for any time inconsistencies, and we ensured that no variables from a future point in time (compared to the point in time the prediction was made) were used. Six standard evaluation metrics are used to assess the predictions of all the classifiers. These include ROC-AUC, PR-AUC, Accuracy, Precision, Recall, and F1 score. ROC-AUC (see figure 1) reflects the best balance between Sensitivity and Specificity whereas PR-AUC (see figure 2) the balance between precision and recall. Ten-fold cross validation was used to avoid overfitting. For training and internal validation the dataset underwent the standard 70% - 30% split. The split was made before the balancing technique was applied.

| Metric / Model | Accuracy | Precision | Recall | F1 |
|---|---|---|---|---|
| **Random Forest** | 0.86 | 0.88 | 0.90 | 0.86 |
| **SVM** | 0.69 | 1.00 | 0.07 | 0.12 |
| **Logistic Regression** | 0.90 | 0.92 | 0.84 | 0.86 |
| **K-Nearest Neighbors** | 0.67 | 0.54 | 0.29 | 0.37 |
| **Decision Tree** | 0.82 | 0.88 | 0.77 | 0.78 |
| **Gradient Boosting** | 0.85 | 0.88 | 0.87 | 0.84 |
| **AdaBoost** | 0.85 | 0.88 | 0.86 | 0.83 |
| **XGBoost** | 0.86 | 0.89 | 0.97 | 0.84 |
| **Naive Bayes** | 0.79 | 0.90 | 0.42 | 0.55 |
| **Bagging** | 0.85 | 0.88 | 0.86 | 0.83 |
| **Optimized Voting** | 0.94 | 0.90 | 0.97 | 0.91 |

TABLE III
PERFORMANCE OF ALL CONSIDERED MACHINE LEARNING MODELS ACROSS SIX EVALUATION METRICS: ACCURACY, PRECISION, RECALL, F1 SCORE, ROC-AUC, AND PR-AUC.

In Figure 3 we can see the global feature ranking for the predictions of the model. After considering the global explanation for the predicted outcome of PTB in pregnancies, it is also essential to comprehend the output of the models

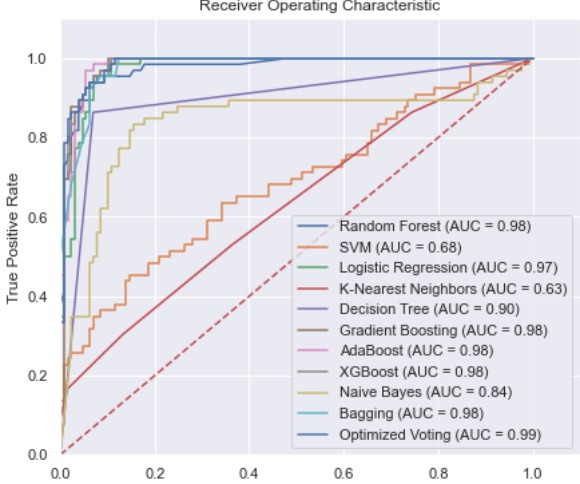
Fig. 1. ROC-AUC for all Classifiers.

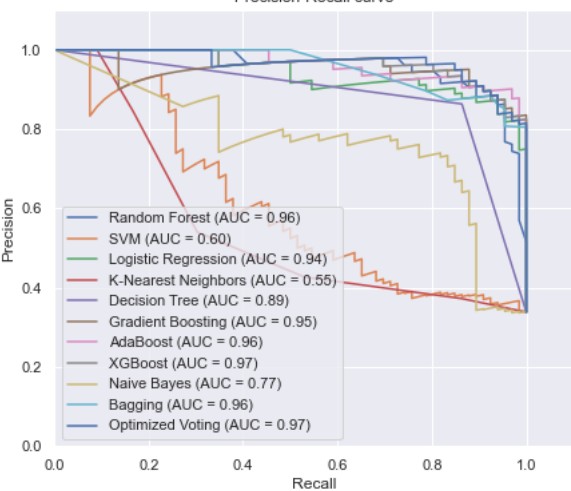
Fig. 2. PR-AUC for all Classifiers.

for each specific case. Figure 4 illustrates an example of the SHAP explanation for one instance randomly selected from the preterm dataset. This output instance was predicted and confirmed as preterm. Here, cervical length and BMI play the most important role in the model's output for this instance. It is worth noting that local and global explanations can differ as presented in this instance. We note here that the average (Global) ranking, and several individual (local) rankings have been checked and confirmed by the medical team.

## IV. CADXPERT OB-GYN APPLICATION

Once the most efficient predictive model (i.e., Optimized Voting) was selected and properly configured, the web application was designed and implemented. It brings together the diverse capabilities of several technology platforms including Keycloak, Angular, HAPI FHIR, Python, and Nginx. Each of these components contributes the proper features to create an integrated system that implements the set requirements.

The architecture is organized into multiple modules, each addressing distinct functionalities within the system. Starting from User Authentication and Authorization (UAA), to the

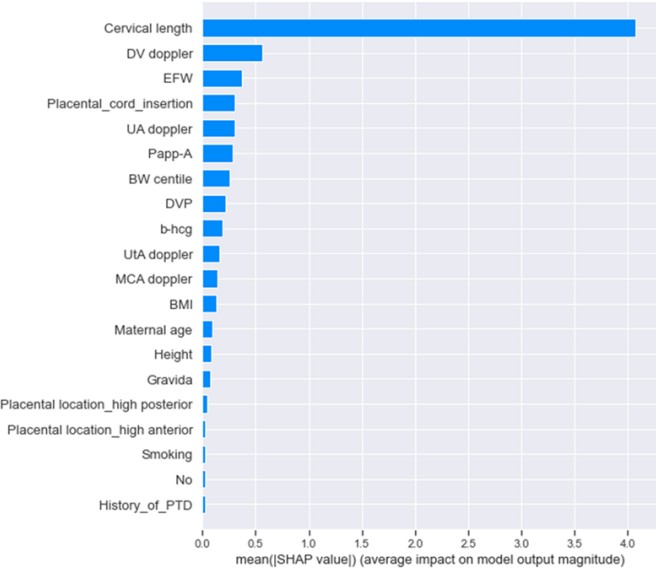

Fig. 3. [Global ranking] depicts the average influence of features on the constructed classifier.

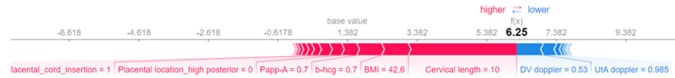

Fig. 4. [Local ranking] depicts feature influence on one randomly selected prediction of confirmed preterm birth risk.

front-end application where data input and management occur, to the Healthcare Data Management Backend, responsible for secure data storage and retrieval, to the Predictive Analytics Application handling outcome predictions, and finally to Networking and Load Balancing which ensures optimal performance and resource allocation.

The combined workings of these components manifest in a user-friendly, highly secure and performance-optimized application capable of delivering key insights to its users, that aims to bring value to healthcare providers by allowing for data-informed decisions, which can lead to more effective treatments and improved newborn outcomes.

In what follows, we will focus in the architectural elements, exploring how they contribute to the system's overall function and impact. This detailed understanding of the system architecture (see figure 5) will provide insight into the intricate workings of this healthcare solution and illustrate the level of sophistication achieved in its design and implementation.

User Authentication and Authorization: The initial layer of the architecture involves User Authentication and Authorization (UAA), managed by a Keycloak instance. It integrates with the other modules of the architecture, offering a standardized, secure method for user management. Keycloak is backed by a PostgreSQL database, a powerful, open-source object-relational database system. It stores user identity data, session information, and other crucial data necessary for effective user management, ensuring a secure and seamless user experience.

Front-end Application: The front-end application is crafted

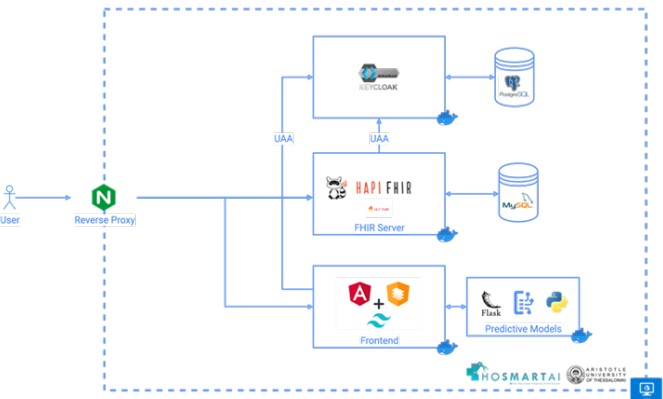

Fig. 5. Software Architecture and System Description.

using Angular, a powerful framework for building dynamic web applications. For building the User Interface, we leverage Material, a design system created by Google, and Tailwind, a highly customizable, low-level CSS framework. The Angular app integrates with Keycloak for user authentication and communicates with the HAPI FHIR[2] server and the predictive models application, to manage and process patient data.

Healthcare Data Management Backend: The healthcare data management component of the architecture is centered around the HAPI FHIR server. FHIR[3] (Fast Healthcare Interoperability Resources) is a standard for exchanging healthcare information electronically, ensuring interoperability between healthcare systems. The HAPI FHIR server integrates with Keycloak, using it for secure user authentication and data access control. The server is backed by a MySQL database, for storing patient data related to Observations and Conditions. The Angular application acts as the conduit through which users can input, view, and edit this data, promoting data accessibility and effective patient management.

Predictive Models Application: The predictive model's application component is a Python application utilizing Flask. The Python app exposes a Flask API, serving as the communication bridge between the predictive models and the Angular application. The API allows users to send patient data from the Angular app, which the predictive models then process to return predictions, enhancing the application's value by enabling more informed healthcare decisions.

Importantly, all patient data managed by the system is anonymized, ensuring compliance with privacy standards and data protection regulations. Anonymization is achieved by replacing identifiable information with non-identifying equivalents, with patients referenced only by an ID.

## V. Feasibility Study Results

In this section the evaluation of the predictive model and its application in a clinical setting is described.

---

[2]https://hapifhir.io/
[3]https://www.hl7.org/fhir/

| Variable | No/Nan | Yes |
|---|---|---|
| **Smoking** | 86 | 14 |
| *Conception_ART* | 92 | 2 |
| *History_of_SGA_FGR* | 97 | 3 |
| *History_of_PTD* | 97 | 3 |
| **Preterm_birth** | 65 | 35 |
| Gravida | 50 | 50 (various 1-5) |
| *Parity* | 67 | 33 (various 1-4) |
| *Placental Cord Insertion* | 27 | 73 (various 1-3) |
| *Placental Cord Insertion Abnormal* | 92 | 8 |
| *Placental Location* | N/A | Observations (Various Locations) |
| *Single Umbilical Artery* | 100 | 0 |
| **Maternal Disease** | 97 | 3 Total (1's from all diseases) |

TABLE IV

LIST OF CATEGORICAL VARIABLES AND THEIR DISTRIBUTION INCLUDED IN THE EXTERNAL VALIDATION'S DATASET.

| Variable | Mean | Std. Dev |
|---|---|---|
| *Maternal age* | 31.95 | 4.80 |
| *BW centile* | 42.85 | 30.36 |
| **UtA Doppler** | 0.80 | 0.23 |
| *b-hcg* | 1.01 | 0.28 |
| **DVP** | 4.80 | 0.92 |
| **MCA Doppler** | 2.02 | 0.23 |
| **Papp-A** | 0.85 | 0.23 |
| *Height* | 166.23 | 6.28 |
| **UA Doppler** | 0.93 | 0.12 |
| *DV Doppler* | 0.58 | 0.19 |
| **EFW** | 1932.49 | 229.80 |
| **Cervical length** | 27.16 | 7.79 |
| **BMI** | 28.76 | 5.04 |

TABLE V

OVERVIEW OF NUMERICAL VARIABLES – LIST OF THE NUMERICAL VARIABLES INCLUDING THEIR MEAN VALUES AND STANDARD DEVIATIONS INCLUDED IN THE EXTERNAL VALIDATION'S DATASET.

### A. External validation of the predictive model

Following the technical development and the successful completion of technical reliability evaluations, collaborating medical personnel from the Hippokration General Hospital of Thessaloniki employed the CADXpert OB-GYN application in real-life clinical settings during pilot tests. The objective was to evaluate the model's performance on new, unseen pregnancy cases to confirm its generalizability and reliability across different clinical environments.

Pregnant women eligible for enrolment were identified at the partnering healthcare facility of the hospital. Enrolment and data collection occurred on the same day, following informed consent. Participants were assigned a unique ID, along with data on demographics, medical and social history, and specific pregnancy-related variables. Data was acquired from 100 participants (data collection participants) as part of routine clinical practice in Hippokration General Hospital. Four maternal-fetal medicine specialists were involved in the study, who provided the ground truth.

Subjects eligible for data collection met the following criteria: 1) Pregnant women with symptoms of preterm labor. 2) Pregnant women with symptoms of Fetal Growth Restriction. 3) Age at least 18 years. 4) Singleton pregnancies. Whereas, subjects were excluded from data collection based on the following criteria: 1) Women who refuse to give written consent to participate in the study. 2) Pregnancies with prenatally diagnosed congenital abnormalities.

The application has certain variables set as required for model prediction and others as optional (based on their importance rank order for the prediction task). Optional variables were collected according to the medical team's judgement per each case, whereas the required ones were collected from all participants. These values along with detailed baseline characteristics are summarized in Table IV and Table V. Optional variables are noted in italics.

The medical team used the application with the 100 participants that were enrolled during the pilot. The application provided the prediction along with the explanation for each prediction. Then, the participants' continued care was provided according to standard clinical practices independent of the CADXpert OB-GYN results. The medical team was then provided with the final results from each case and collected them along with the results from CADXpert OB-GYN. Example executions are depicted in figure 6 where the variables are being entered by the clincial, and in figure 7 where the result is presented and explained to the user. These results were then used to evaluate the application.

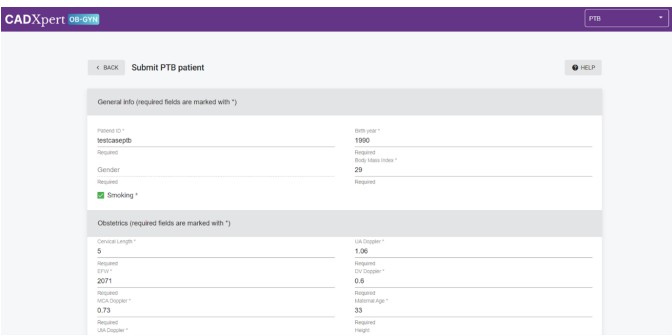

Fig. 6. Data entry for a potential pre- term birth case.

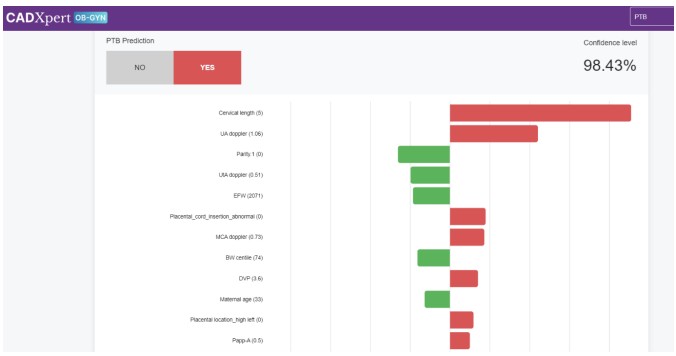

Fig. 7. Prediction and explanation for a case predicted as of high risk for preterm birth. This woman is predicted to deliver preterm with a confidence level of 98.43% and the contribution of each variable to the prediction is described below.

The model's performance was evaluated using the same metrics as in the internal validation, reflecting its predictive capabilities in a real-world clinical setting. Specifically, it achieved accuracy = 0.890, precision = 0.786, recall = 0.943,

F1 score = 0.857, ROC AUC = 0,902 and PR AUC = 0.891. Two important metrics are the Precision-Recall (PR) Curve and the Receiver Operating Characteristic (ROC) Curve. The first curve (see figure 8) shows the trade-off between precision and recall for the model, with a focus on its utility in a clinical scenario where high recall is critical. The area under the curve (AUC) of 0.891 suggests a good level of performance, especially in scenarios where the positive class is rare. The second curve (see figure 9) illustrates the model's ability to distinguish between classes effectively. The ROC curve demonstrates the model's discrimination capacity. With an AUC of 0.902, the model has a good ability to differentiate between the positive and negative cases.

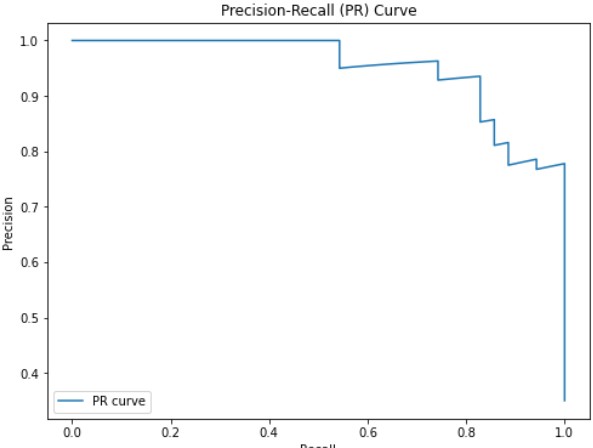

Fig. 8. Precision-Recall (PR) Curve

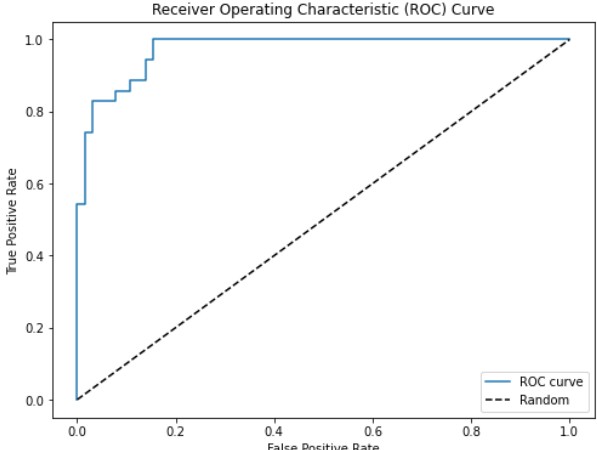

Fig. 9. Receiver Operating Characteristic (ROC) Curve.

### B. Evaluation of user satisfaction

Following the pilot deployment of the application, the medical team that used it subsequently provided evaluations of its performance. A total of 10 obstetricians participated in these tests, with the sample comprising an equal number of male (5/10) and female (5/10) physicians. The participating physicians had varying years of experience, ranging from approximately 1 to 12 years, and were ensured sufficient system usage over the course of 8 months to enable objective evaluations. Upon completion of the pilots, the collaborating obstetricians were asked to evaluate their experience using validated questionnaires. Specifically, the Usefulness, Satisfaction, and Ease of Use (USE) questionnaire was employed to assess perceived usefulness, while the System Usability Scale (SUS) questionnaire was used to evaluate perceived usability of the CADXpert OB-GYN application.

Regarding usability, 40% of participants rated the application as D – Poor, another 40% rated it as B – Good, and 20% rated it as A – Excellent. The average score on the SUS questionnaire was 71.50, corresponding to a grade of B – Good, with 50% of the scores slightly exceeding the average (72.50). As for usefulness, over 80% of participants responded positively to each question in this category, confirming the application's utility in daily medical practice. Concerning ease of use, the majority of users, with few exceptions, agreed to varying extents that the application is simple, flexible, easy to use, and features a user-friendly environment with minimal steps required to complete actions. However, regarding ease of learning, slightly over 70% responded positively, indicating a need for a user manual to facilitate future users. Finally, in terms of satisfaction, over 80% of participants indicated varying levels of agreement with each question, suggesting satisfactory levels of user satisfaction.

Regarding the correlation between the evaluators' years of experience and their evaluation results, a clear trend was observed, indicating that increased years of experience are associated with higher ratings. This outcome is expected given that the current incident assessment process predominantly relies on the expertise of the medical staff.

To assess the reliability of the responses, a Cronbach's Alpha analysis was conducted. The alpha coefficient for all USE categories exceeded 0.9, indicating excellent internal consistency. Conversely, the alpha coefficient for the SUS was 0.37, reflecting poor consistency. This discrepancy may be attributed to limited variability in responses or the homogeneous characteristics of the participants.

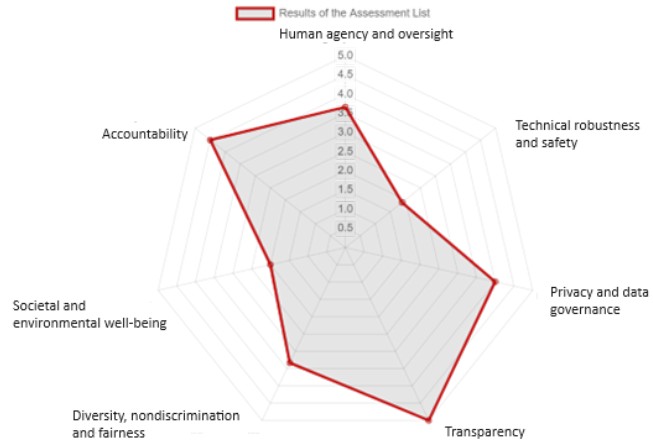

Fig. 10. ALTAI assessment results - summary.

Finally, CADXpert OB-GYN was evaluated based on the Assessment List for Trustworthy Artificial Intelligence (AL-

TAI),[4] showcasing (see Figure 10) strong performance in transparency, data privacy and accountability, and fair to good performance in the rest of the metrics.

## VI. CONCLUSIONS AND FUTURE WORK

This study demonstrates the effectiveness of our AI-based decision support tool in predicting preterm birth, with both internal and external validations showcasing strong model performance. Compared to the existing state-of-the-art our approach demonstrates equal or superior performance with higher accuracy and recall rates in both internal and external validations and includes the development of a web-based application for clinical routine use, emphasizing usability and transparency in predictions using an explainable XAI technique. The use of SHAP explanations further enhances the interpretability of the predictions, making the tool more trustworthy and usable for clinicians. This is the first research work that aims to evaluate a clinical predictive model regarding premature birth risk based on its accuracy, interpretability and clinical usability, ensuring that it may become applicable in the clinical practice. This tool can be particularly useful for clinicians to identify pregnancies in risk of preterm delivery and thus introduce early intervention strategies, something highly important especially in resource limited environments.

There are a few limitations to note. The external validation involved a small sample of 100 patients from a single clinical setting. Although the results were promising, larger studies across diverse populations and settings are needed to confirm the model's generalizability. Additionally, while the model predicts preterm birth likelihood, it does not estimate the exact week or date, which could enhance clinical decision-making.

Future work will expand the dataset and conduct broader validation studies to refine the model. Additionally, the feasibility of predicting the specific gestational age at delivery will be explored to enhance the tool's clinical utility. Finally, the applicability of the findings of this work in other health-related domains such as the prediction of the outcome of cancer treatments (e.g., urologic cancers) will be investigated.[5]

## AKNOWLEDGEMENT

This work was co-funded by the European Union's Horizon 2020 Innovation Action programme, 'HosmartAI - Hospital Smart Development Based on AI,' under Grant Agreement No. 101016834, and by the European Union's Horizon Europe Research and Innovation Work Programme, 'COMFORT - Improving Urologic Cancer Care with Artificial Intelligence Solutions,' under Grant Agreement No. 101079894.

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
