# OpenReview forum: "An Explainable AI-Based Decision Support Tool to Predict Preterm Birth"
_IEEE.org/EMBS/BHI/2024/Conference — IEEE BHI'24_

### Official Review · Reviewer_qSV8 · 2024-08-03
**An Explainable AI-Based Decision Support Tool to Predict Preterm Birth**

**Overall Rating:** 8
**Confidence:** 5

**Other Quality Metrics:**

(a) Clarity of Writing: Good
(b) Clinical Significance: Great
(c) Methodological Novelty: Good
(d) Experiments and Results: Excellent

**Questions For The Authors:**

N/a

**Strengths:**

The work presents promising developments in the use of AI to predict preterm delivery, with several key strengths that could significantly improve clinical practice The machine learning model demonstrated high prediction accuracy, reaching 94%; accuracy and 97% residual in internal validation The feature is the integration of of SHapley Additive explanations (SHAP), which adds greater explanatory power to the model’s predictions, and allows physicians to understand and trust AI decisions—a key factor which makes it acceptable in terms of health.

Using the prototype in a user-friendly web application further increases its potential impact. The application is designed to be secure, flexible, and easily integrated into clinical practice, reducing the need for extensive training. External validation of the model also took place in 100 pregnant women in a real-world clinical setting, where it performed consistently well, demonstrating its generalizability and robustness across different settings.

Positive feedback from clinicians who used the application in the pilot study highlights the practical benefits, especially the benefits of the translation aspect. This compatibility with the needs of the user suggests that the tool can be widely used in clinical settings. Furthermore, the tool’s ability to improve antenatal care, particularly in remote or underserved settings, highlights its broad impact, including potential capacity reduction neonatal morbidity and mortality by predicting the initial prognosis of preterm birth.

**Summary Of The Paper:**

The paper presents the development of an AI-based clinical decision support system (CDSS) that is interpretable for predicting preterm birth (PTB), defined as birth before 37 weeks of gestation When the authors used a dataset of 873 pregnancies to test machine learning models , because with excellent performance, the model achieved high accuracy (94%) and recall (97%) in internal validation, and departed resulted in strong external validation in 100 pregnant women (89% accuracy and 94.3% recall).

An important feature of this system is the SHapley Additive exPlanations (SHAP) integration, which streamlines model interpretation, makes it easier for clinicians to understand predictions and gain confidence The model has been implemented in a web application which easy to use designed to be used effectively and efficiently in clinical settings.

Clinicians participating in the study provided positive feedback, particularly valuing the interpretation and usefulness of the tool. The authors suggest that future work could include expanding the dataset and conducting larger clinical trials to further validate the model. Overall, the paper demonstrates the potential of integrating AI that can be translated into clinical practice for better prognosis and management of PTB.

**Weaknesses:**

Although the paper provides a robust and promising tool for predicting preterm birth, a minor weakness in the limited external validation is that although the model was tested in pregnant women 100 in a real-world clinical setting though that sample size is relatively small. Larger studies in different populations and clinical settings would be useful to establish reliability and generalizability so in the pattern that may be present in different health care settings is fully emphasized. Extending the external validation could help to improve the consistent performance of the model in a wider range of settings, thereby increasing its reliability and potential for widespread validation.

---

### Official Review · Reviewer_qtcS · 2024-08-10
**An Explainable AI-Based Decision Support Tool to Predict Preterm Birth**

**Overall Rating:** 7
**Confidence:** 2

**Other Quality Metrics:**

Clarity of Writing: Great
Clinical Significance: Good
Methodological Novelty: Good
Experiments and Results: Great

**Questions For The Authors:**

- Which of the models ended up in the application?
- How far into the future does the model predict?
- Would it be possible to predict the date (or week) of birth?

**Strengths:**

- Good preparation of used data
- Vast variety of implemented methods
- Good results and evaluation

**Summary Of The Paper:**

The authors developed an online support tool, which helps clinicans to predict pre-term birth.
They implemented, tested and evaluated several alogirthms as candidates for the application.
Several of the algorihms showed promising results.
A live experiment, where the application was testes in clinical environment, was able to reproduce the internal results.

**Weaknesses:**

- I am missing a neural network in the comparing methods
- Better outline the potential use and benefits of the application

---

### Official Review · Reviewer_26nT · 2024-08-15
**The results are good but the figures and the tables lack clarity**

**Overall Rating:** 4
**Confidence:** 4

**Other Quality Metrics:**

Clarity - Poor
Clinical Significance - Good
Methodological Novelty - Fair
Experiments and Results - Good

**Questions For The Authors:**

1. Among the input features used, how did the authors ensure that the features from the future (with respect to the point when the prediction is made) are not being used?
2. How did the authors split the training, validation, and test sets? Did they separate the test set before or after balancing the data?

**Strengths:**

The paper uses 11 different existing solutions and presents a comparison of their performances in terms of accuracy, precision, recall, and F1 scores. The best solution achieves an accuracy of 0.94 and an F1 score of 0.91, which are often difficult to achieve in clinical datasets.

**Summary Of The Paper:**

This paper addresses the important problem of predicting preterm birth in advance with existing machine learning (ML) techniques. It also has an explainable AI component that presents the SHAP values for the predictions.

**Weaknesses:**

Some of the figures in this paper have very low quality, so they do not convey any useful information to the readers. The authors should use higher resolution figures and larger and bold fonts for the texts within the figures. In general, the results lack proper discussion and the table captions are too short to help the readers understand the significance.

---

### Decision · Program_Chairs · 2024-09-23

Accept